

# Introducing the municipal digital offering index for evaluating online services and addressing the digital divide

Carolina Busco, Felipe González, Paula Farina, Jonathan Vivas, Fernanda Saavedra and Lizbeth Avalos

Faculty of Industrial Engineering, Universidad Diego Portales, Santiago, Chile

## ABSTRACT

This research introduces the Municipal Digital Offering Index (MDOI) to assess municipal online service development in Chile. The study utilizes content analysis of municipal websites, creating a systematic instrument to evaluate digital services. It evaluates all 344 Chilean municipalities based on 163 dichotomous variables. Through factor analysis and regression modeling, it investigates sociodemographic and economic factors influencing digital development at the municipal level, offering insights into the digital divide across municipalities. The findings highlight geographical disparities and indicate priority intervention areas. While education levels and financial resources influence digital technology adoption, many municipalities lack efficient online procedures, prompting focused digital transformation investments. This research emphasizes the importance of localized digital services in bridging the digital divide and promoting inclusive governance.

## INTRODUCTION

Contemporary society is experiencing significant changes, notably underscored by the pandemic. The emergence of the fourth industrial revolution, representing a novel production paradigm, instills shifts originating from manufacturing but extending throughout society (*Hobsbawm, 2011*; *Oztemel & Gursev, 2020*). This transformation is characterized by the fusion of technologies across physical, digital, and biological realms (*Schwab, 2017*).

Information and communication technologies (ICT) have assumed pivotal roles in the public and private sectors. Governments have endeavored to transform electronic administration to enhance efficiency, transparency, and citizen-state interaction (*Bayona & Morales, 2017*; *Ndou, 2004*). In the digital age, citizens expect seamless access to public services around the clock from any location, necessitating a well-structured approach to delivering personalized services (*Leach et al., 2021*).

To meet these expectations, the concept of e-government arose. According to *The World Bank (2015)*, e-government involves utilizing ICT to enhance how the government operates, making it more efficient, effective, transparent, and accountable. *Silcock (2001)*, on the other hand, sees it as leveraging technology to make government services more

Corresponding author
Paula Farina,
paula.farina@mail.udp.cl

accessible and efficient, thereby benefiting citizens, businesses, and government employees while fostering a closer relationship between the government and the public. Additionally, digital government emphasizes delivering benefits to citizens through ICT, often involving collaborative efforts among different entities (*Hashmi et al., 2018*).

Chile began its journey into e-government in 1999 with the establishment of the Interministerial Committee for the Modernization of Public Management. Since then, various initiatives have been implemented to modernize the public sector, such as launching the first state portal and crafting the government's digital agenda titled 'Chile Towards the Information Society'. These endeavors illustrate Chile's ongoing commitment to adapting to the digital era and enhancing communication between citizens and the government (*Gutiérrez Campos, 2019*).

Despite relative advances in digitalization and e-government in Chile, according to international indicators (*CISCO, 2019*; *Ubaldi & Okubo, 2020*), significant challenges persist for proper implementation, including socioeconomic inequality, hierarchical culture, and lack of adequate education. Based on the above, this research addresses two research questions:

RQ1: What is the development level of the municipal digital offering in Chile?

RQ2: What demographic, social, and economic factors are related to this development level?

To address questions 1 and 2 from a quantitative perspective, two main objectives are established: (i) Constructing an index, named the Municipal Digital Offering Index (MDOI), to measure and compare the advancement of municipal online services; (ii) Investigating social and economic factors that contribute to the MDOI in Chilean municipalities. The findings, broken down into multiple dimensions and geographically contextualized, offer a global perspective on the current state of digital implementation in Chilean municipalities. Nevertheless, this methodology is easily reproducible to evaluate any digital offer at the local government level in the world.

The methodological process begins with creating a content analysis instrument to observe and quantify relevant variables on the websites of Chilean municipalities. Collected data is then analyzed using exploratory factor analysis to identify key dimensions and simplify the dataset. Subsequently, an index is created to evaluate the level of development of municipal websites. Finally, regression analyses are employed to assess how sociodemographic and economic variables influence the MDOI.

The motivation for conducting this research stems from the frequent interactions between citizens and local governments, which are more common compared to interactions with national authorities, making the provision of online services at the local level essential. On the other hand, according to the *United Nations (2022)*, the most effective way to improve e-government is to regularly assess and evaluate government portals, which must be able to accommodate larger numbers of users.

Municipal websites are often dynamic and interactive, although they tend to be less sophisticated than national portals (*Bayona & Morales, 2017*). On the other hand,

municipalities, as local centers of governance, reflect their residents' social and economic diversity. Therefore, assessing the development level of their websites can serve as a valuable indicator of the digital divide throughout the country.

While e-government indexes are periodically applied, they mostly target the national level rather than the municipal level. Examples include the Digital Government Index (*OECD, 2024*) and the Electronic Government Development Index (*United Nations, 2022*). In contrast, existing municipal-level indices, such as the Local Online Services Index (*United Nations, 2022*), focus on capital cities, which are urbanized areas representing central policies. MDOI is designed to assess the level of digital development reflected in the online service offerings of municipalities, including rural districts. In this context, digital development refers to the degree to which a municipality adopts, implements, and enhances digital technologies to provide public services online, engage with citizens, and manage internal operations. Rather than evaluating digital transformation in abstract or organizational terms, the MDOI focuses specifically on the tangible development of municipal digital service provision.

To assess digital development at the municipal level, the MDOI covers seven dimensions: (1) municipal data—gathers comprehensive information on the general characteristics and socio-economic indicators of each municipality; (2) information content—encompasses various types of information aimed at informing, educating, and engaging residents; (3) bidirectional transactions—evaluates the extent to which municipal websites enable users to efficiently complete digital procedures and access dynamic information; (4) interaction—assesses communication tools and opportunities for citizen participation provided through digital channels; (5) integration—examines the municipality's efforts to ensure equitable access and support for all communities, particularly underserved groups; (6) e-democracy—focuses on the availability of digital tools and resources that promote democratic practices and citizen empowerment; and (7) security—addresses user safety by evaluating proactive measures, available resources, and coordination with relevant stakeholders to tackle digital security challenges and protect citizens' well-being. By quantifying these dimensions, the MDOI offers a comprehensive view of how developed a municipality's digital public service offerings are.

The remaining sections of this text are structured as follows: 'Literature Review' presents the literature review, followed by the research methodology used ('Materials and Methods'). Subsequently, 'Results and Discussion' presents the findings, along with their respective analysis and discussion. Finally, 'Conclusions' presents the conclusions drawn from the results.

## LITERATURE REVIEW

The analysis of municipal digital services in Chile constitutes an ever-evolving field of research, gaining increasing relevance in the context of the modernization of public administration. This literature review provides a comprehensive overview, contributing to

understanding the conceptual definitions of e-government, the international indices before MDOI, and the specific laws governing municipalities in Chile. These three issues were considered when creating MDOI.

## Conceptual evolution and impact of e-government

E-government refers to the use of technology to enhance access to and delivery of government services, benefiting citizens, businesses, and employees. It aims to create a modernized, integrated, and transparent public service model, fostering collaboration between governments and citizens (*Silcock, 2001*). Over time, e-government has evolved into digital government, maintaining the core principles of the original definition. As research progresses, the concepts become more diverse and perspectives more intricate. For instance, differences in the pace of this transformation between national/federal governments and local governments are noticeable (*Manoharan & Ingrams, 2018*). Additionally, studies explore the role of digital government in promoting transparency, trust in government, and combating corruption (*Arayankalam, Khan & Krishnan, 2021*; *Tolbert & Mossberger, 2006*), its impact on specific social groups (*Hien, 2014*), and the influence of social media in its management (*Kahn et al., 2013*).

Over the past 20 years, e-government studies have focused on assessing the extent and quality of services provided through ICTs (*Mensah, Zeng & Luo, 2020*). Researchers have also developed maturity models to aid in planning modernization strategies. Various obstacles have been identified, including the digital skills of citizens, the establishment of informative platforms and streamlined processes, and cybersecurity concerns (*Das, Singh & Joseph, 2017*; *Davidson, Wagner & Ma, 2015*). To analyze e-government, indicators can be categorized into four main areas: government-to-citizen (G2C), government-to-business (G2B), government-to-employee (G2E), and government-to-government (G2G). The latter involves communication between different government agencies, often utilizing extensive databases to promote interoperability (*Alshehri & Drew, 2010*).

At the national level, according to *Reffat (2003)*, e-government represents a significant shift in how the government interacts with the public, beyond just setting up computers or websites; instead, it entails using technology to improve how government services are provided. The process of implementing e-government involves five stages: deciding to make changes in systems, which is influenced by both internal and external factors; spreading these changes, which depend on background and knowledge; institutionalizing e-government practices; and evaluating the extent to which these changes have been put into practice (*Dias, 2020*).

*Ashaye & Irani (2019)* introduced a series of developmental stages illustrating the gradual improvement in the interaction between government and citizens through digital tools. The first stage, transmission, focuses on providing government information through websites and other electronic means. Next is the interaction stage, where there is increased two-way communication, allowing citizens to ask questions and give feedback *via* digital platforms. The third stage is a transaction, which marks the point where electronic transactions like tax payments and online form submissions become possible. Finally, the integration stage, the most advanced phase, involves fully incorporating digital

technologies to organize and manage various government services comprehensively and efficiently. These stages depict the progression of a government towards offering more advanced and efficient electronic services.

Nam (2014) identified several ways in which e-government is utilized. This encompasses transactional services, where electronic platforms facilitate government-related transactions like online payments. Citizens also utilize electronic resources to access general governmental information and obtain details on government policies through digital platforms. Furthermore, citizen participation involves engaging in online interaction with the government to take part in decision-making processes and offer feedback. Lastly, policy co-creation involves actively collaborating with the government in formulating and developing public policies.

In the same study, Nam (2014) identified five key factors that affect how citizens adopt and use e-government services. These factors are psychological predisposition, trust in the government, information channels, demographic conditions, and civic mindset. It is important for citizens to effectively utilize government electronic services. Additionally, Nam (2014) found that e-government users can be categorized into two main groups: policy researchers and citizens who use digital channels to engage with the government.

Malodia et al. (2021) indicate that the success of e-government is influenced by factors like citizen orientation, channel orientation, and technology. Effective e-government implementation brings about tangible benefits and improves the efficiency of government operations. Nevertheless, challenges such as the digital skills gap and economic inequality can weaken the relationship between e-government and citizens. Additionally, a perception of government transparency positively impacts the less tangible outcomes of e-government.

A relevant line of research focused on benchmarking e-government: Skargren (2020), Przeybilovicz, Cunha & Ribeiro (2023). This literature highlights the importance of continuously and critically examining how e-government development is defined and proposes an effective benchmark that must be sensitive to regional and local contexts, as well as aligned with the core purposes of public administration—namely, to deliver better services and generate public value. As such, any evaluation of e-government should move beyond standardized metrics and consider the diverse institutional, social, and cultural conditions in which digital governance is implemented.

## E-government comparative rankings

In e-government, transparency plays a pivotal role in fostering accountability, trust, and effective governance (Arayankalam, Khan & Krishnan, 2021; Tolbert & Mossberger, 2006). Transparent practices ensure that citizens have access to information about government actions, decisions, and processes, thus empowering them to hold their leaders accountable and participate meaningfully in democratic processes. Moreover, transparency enhances the efficiency and effectiveness of e-government initiatives by promoting openness and accountability in the delivery of public services (Bayona & Morales, 2017; Ndou, 2004).

To gauge the progress and effectiveness of e-government implementations, various studies conduct rankings that assess the level of development and performance of digital government initiatives worldwide. These rankings serve as tools for policymakers and researchers, offering insights into the strengths, weaknesses, and overall development of e-government systems across different countries.

The Institute of Digital Government has been producing the International Digital Government Ranking since 2002. In the 2024 ranking, Singapore (94.69); the UK (94.49) and Denmark (93.29) lead the index, while Chile holds the 41st position (67.44) (*WASEDA-IAC, 2024*; *International Academy of CIO (IAC), 2022*).

Conversely, the OECD Digital Government Index 2023 (*OECD, 2024*) assesses how well countries implement data and digital technologies in various aspects, such as digital design, government operations as a platform, data-driven decision-making, transparency, user-centric approach, and proactivity. According to the Digital Government Index (DGI) findings, many OECD countries faced challenges due to insufficiently trained personnel with digital expertise. In this context, Korea, Denmark, and the United Kingdom lead this ranking with scores exceeding 0.778, while Chile ranks 32$^{nd}$ with an index of 0.398 as one of the last places.

Additionally, the United Nations E-Government Development Index (EGDI) (*United Nations, 2022*, *2024*) provides a global benchmark for assessing the extent of digitalization in national governments. The index combines the Online Service Index, the Telecommunication Infrastructure Index, and the Human Capital Index. While 2022 results highlighted advancements in global e-government adoption, with more countries enhancing their EGDI scores, the last results for 2024 show a significant narrowing of the digital divide: the proportion of the world population lagging in digital government development fell from 45.0% to 22.4%. This progress is largely driven by gains in Asia, notably the rise of India and Bangladesh above the global average. The Americas also demonstrated steady advancement, with a greater number of countries joining the very high EGDI group. While Africa and Oceania have made some progress, they continue to lag behind the global average. These results reflect a global trend toward more inclusive digital governance, though regional disparities persist.

The comparison between studies reveals differences in their emphasis: For example, the DGI focuses on aspects of government digitalization, whereas the EGDI takes a broader view, considering both human capacities and telecommunication infrastructure. However, due to different measurement methods and country samples, e-Government indicators remain inconsistent across rankings. This discrepancy poses challenges in generating coherent impressions and highlights the need for standardized methods to facilitate research in this field.

*Dias (2020)* suggests that socioeconomic factors and local political aspects impact how e-government is implemented in municipalities. Evaluation frameworks, such as the one proposed by *Tejedo-Romero et al. (2022)*, consider important factors like online presence, access to urban information, interaction, transactions, and e-democracy to gauge the progress of digital services at the municipal level. Transparency is identified as a vital element for the effective management of resources in local governments, and citizen

involvement is closely tied to electronic transparency, which relies on the availability of information to encourage active participation.

The United Nations regularly assesses local e-government development through the Local Online Services Index (LOSI), a tool designed to evaluate municipal digital portals worldwide. These evaluations are increasingly relevant as urban populations grow and internet access becomes more widespread. In 2024, LOSI underwent a significant methodological update, expanding from 86 to 95 indicators and from five to six dimensions. This revision responds to the need for more inclusive and context-sensitive assessments of digital government, in line with global trends in public engagement and digital inclusion.

The six dimensions in the 2024 LOSI are: (1) institutional framework (five indicators), covering municipal digital strategies, governance structures, legal frameworks on access and privacy, and open data policies; (2) content provision (30 indicators), assessing the availability of essential public information online; (3) services provision (30 indicators), evaluating the accessibility and delivery of government services; (4) participation and engagement (10 indicators), measuring citizen interaction and participatory mechanisms; (5) technology (18 indicators), focusing on portal accessibility, usability, and technical standards; and (6) e-government literacy (10 indicators), a newly introduced dimension measuring the population's ability—especially among vulnerable groups—to effectively use digital public services.

This updated version aligns LOSI more closely with the national-level Online Service Index (OSI), offering a more coherent view of e-government performance across levels of administration. Notably, the 2024 results show Santiago de Chile ranked 74th globally, while Bogotá leads Latin America in 16th place. European cities such as Madrid and Tallinn also score among the highest, reflecting mature local digital governance ecosystems.

To further clarify the originality and positioning of this study within the existing literature, Table 1 summarizes key contributions, methodological approaches, and gaps identified across the main strands reviewed. The table also highlights how MDOI responds to these gaps by offering a context-sensitive, operational tool tailored to the Chilean municipal landscape.

## Legal and regulatory framework of e-government in Chile

A legal framework is crucial for examining how e-government is implemented and operates within municipalities, as well as for assessing their adherence to transparency and information access standards. In Chile, the Law on Digital Transformation of the State (Law No. 21,180: *Government of Chile, 2019*) was published in November 2019 and came into force in 2021. This modifies the basis of the administrative procedures for its transformation and digitalization, promoting digital requests with the support of interoperability, for better integration between public institutions.

Chile's territory is presently divided into 16 regions for the State's governance and internal management. Each region is further divided into 56 provinces. These provinces are then subdivided into 345 communes for local administration purposes. The legal and

**Table 1  Summary of research and indexes on e-government.**

| | Theme | Source | Level & coverage | Method/Metric | Main contribution | Identified gap | How MDOI responds |
|---|---|---|---|---|---|---|---|
| 1 | Conceptual evolution. | Silcock (2001) | Conceptual, international | Foundational definition of e-government. | Establishes service-improvement focus. | Lacks operationalized, local-level frameworks. | Provides an applied, municipal-level operationalization of digital development. |
| 2 | Conceptual evolution. | Nam (2014) | Empirical, international | Typology of e-gov use & adoption factors. | Links citizen trust & usage patterns. | Focuses on demand-side usage, and lacks service supply indicators. | Captures digital service functionalities offered by municipalities. |
| 3 | Benchmarking critique. | Skargren (2020) | Integrative review | Critical examination of e-gov benchmarking. | Argues for context-sensitive benchmarking methods. | Existing benchmarks often ignore the national/local context. | Is designed specifically for Chile's institutional and territorial context. |
| 4 | Decolonizing benchmarking. | Przeybilovicz, Cunha & Ribeiro (2023) | Global South focus | Conceptual & exploratory. | Calls for decolonized and situated benchmarking approaches. | Standard metrics overlook sociocultural realities in the Global South. | Incorporates a context-aware framework aligned with Chilean public sector characteristics. |
| 5 | Comparative rankings. | WASEDA-IAC (2023, 2024) | 85 countries, national | Composite scorecard. | Annual cross-country e-gov ranking. | Applies uniform indicators across diverse nations. | Assesses digital development across all Chilean municipalities, including rural areas. |
| 6 | Comparative rankings. | OECD (2024) | 38 OECD members | Six-pillar index. | Assesses national digital government maturity. | Lacks subnational granularity. | Captures intra-country variation in municipal digital development. |
| 7 | Comparative rankings. | United Nations (2022, 2024) | 193 UN members | Online services, infrastructure, human capital. | Tracks global e-gov progress over time. | Focuses on national level, limited local insight. | Provides subnational granularity and contextual relevance. |
| 8 | Comparative rankings. | United Nations (2022, 2024) | 193 capital cities | 95 indicators, six dimensions. | Measures digital service offerings at the city level. | Excludes smaller cities and rural municipalities. | Includes all 344 Chilean municipalities, regardless of size or location. |
| 9 | Legal/regulatory (Chile). | Government of Chile (2019) | National | Mandates digital transformation and interoperability. | Establishes an institutional framework for digital public administration. | No implementation monitoring mechanism. | Tracks progress in municipal compliance with digital mandates. |
| 10 | Legal/regulatory (Chile). | Government of Chile (2008) | National | Transparency and access to information requirements. | Defines active and passive transparency obligations. | No systematic assessment of municipal website content. | Evaluate the actual availability of digital information and services. |

regulatory framework within Chilean municipalities operates under Law No. 18,695, also referred to as the Organic Constitutional Law of Municipalities.

Regarding transparency and public information accessibility, the Transparency Law, (Law No. 20.285: *Government of Chile, 2008*), approved in 2008, acknowledges everyone's right to access public information and functions through two primary aspects: First, active transparency entails public entities' responsibility to provide pertinent and up-to-date information regularly. Secondly, passive transparency (or the right of access to information) mandates that public bodies respond to any individual's information inquiries.

According to Article 7, municipal entities are required to provide the public with essential background information on their websites, updating it at least once every month, covering details on the organizational structure, responsibilities, functions, and authorities of each unit or internal body, the municipal regulatory framework, staff information, employment practices, public fund transfers, service access procedures and requirements, subsidy initiatives, budget allocation, outcomes of internal audits, and other pertinent data.

## MATERIALS AND METHODS

This study adopts a specific methodological approach to achieve two main objectives: (i) develop an index called the MDOI to assess the progress of municipal online services and allow comparison, and (ii) examine social and economic factors that influence the MDOI in Chilean municipalities. For this purpose, data on the municipality's websites are gathered using content analysis, and an exploratory factor analysis is utilized to create the index. Once the MDOI is established, additional socioeconomic data from the communes is collected to explore its relationship with the MDOI index. Multivariate linear models are employed for this purpose.

### Data gathering

The research design employs a probabilistic, descriptive, empirical, cross-sectional, and quantitative approach. The unit under study is the websites of Chilean municipalities, encompassing the entire population of 344 municipalities. As a research instrument, a content analysis guideline was constructed to quantitatively record (*Kerlinger, 1975*; *Espín, 2002*) the digital services offered by municipalities (see 'Research Instrument' in Supplemental Material). This instrument enables a systematic and reproducible collection of information and facilitates the inference of content, aiding in the establishment of indicators. It consists of a custom-designed tool to evaluate all the websites of Chilean municipalities based on 163 dichotomous variables that address information on the web portals in compliance with the Chilean transparency law. Based on the findings of the literature review, these variables are classified into six main dimensions and 30 sub-dimensions. Dichotomous variables can only have two results: "YES", indicating the presence of the required information, or "NO", indicating its absence. The complete instrument can be found in Table A1, and a summary of the instrument's dimensions is schematized in Fig. 1. The research instrument was applied in 2022 by researchers from Universidad Diego Portales in Chile.

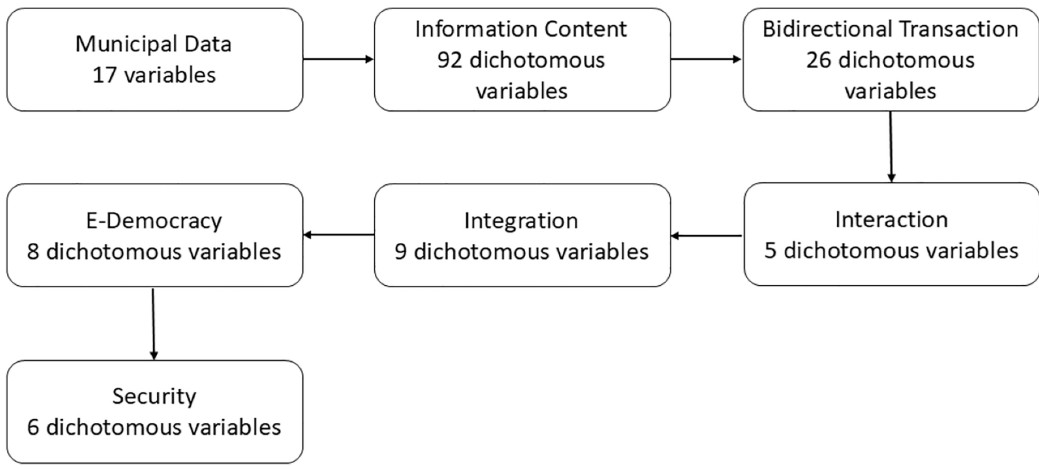

**Figure 1 Structural scheme of the research instrument.**

The information collected from websites is complemented with additional data gathered from official governmental sources like *Electoral Registry of Mexico (INE) (2017)*, *CASEN (2020)*, and *División Observatorio Social MDSF-CEPAL (2022)*. Variables included are Region, Community Development Index (CDI) 2020, Area of the commune in km$^2$ (Size), Population Quantity Poverty Rate %, and Income (from Municipal Patents and Circulation Permits, in Million $).

## Exploratory factor analysis

The exploratory factor analysis (EFA) aims to use the data collected on the municipalities' websites to construct an index able to measure the MDOI. The EFA model operates under the assumption that observable dichotomous variables are linearly connected by unobservable factors. The purpose of EFA is to recognize shared factors that account for the intercorrelations among the observed items, in the sense that when the factors are held fixed, the observed items would be uncorrelated (*Harman, 1960*; *Lloret-Segura et al., 2014*; *MacCallum et al., 2001*). This allows information from numerous variables to be condensed into smaller factors. Within the EFA framework, the number of factors is unknown, and a criterion to decide on is required. Parallel analysis (*Horn, 1965*), jointly with the scree plot derived from it, is employed to decide the number of factors in this study.

Given that our observable variables are dichotomous, the assumption of multivariate normality is not appropriate. We employ the three-stage estimation method (*Jöreskog, 1990*; *Muthén, 1984*) to deal with it. The first step of this method consists of estimating thresholds of underlying response variables, which give the origin of the dichotomous variables. In the second step, given the thresholds previously obtained, the tetrachoric correlation matrix of the data, a dichotomous version of the Pearson correlation matrix, is estimated (*Jöreskog, 1994*). Finally, the resulting tetrachoric correlation is used to estimate

the EFA employing the method of least squares extraction (*Harman, 1960*; *Madrigal-Delgado et al., 2018*). It is well known that the solution for the parameter's estimation procedure is identified up to an orthogonal transformation of the factor matrix. In that sense, a rotation method that facilitates factor interpretation must be chosen. In this study, we employ an oblivion rotation procedure. The criteria for assigning variables to factors consisted of considering factor loadings bigger than 0.4, following the guidelines set by *Lloret-Segura et al. (2014)* and *MacCallum et al. (2001)*. Finally, we use the KMO measure to confirm the suitability of the data for factorization, and Cronbach's alpha coefficient to assess the reliability of the instrument. The R Psych package conducted the EFA model's computational implementation (*Revelle, 2024*).

The factors obtained from the EFA were conceptually analyzed to identify the implicit abstract concept captured by the variables linked to each factor.

## Index creation

The primary aim of this study is to develop a standardized tool called the MDOI. This index serves as a comparative measure for evaluating the digital services offered across various municipalities and facilitates their comparison. To construct the MDOI, we utilize the factors identified in "Exploratory Factor Analysis". First, we estimated the Mean Index from Factor Loadings (MIFL) as defined in Eq. (1)

$$MIFL_{jk} = \left( \frac{\sum_{i=1}^{D_j} X_{ijk} * FL_{ij}}{SFL_j} \right) \ \forall \ k \in \{1, \ 2, \ \ldots, \ 344\} \tag{1}$$

In this equation, $X_{ijk}$ represents the result of a dichotomous variable $i$ related to fact or $j$, for municipality $k$. $FL_{ij}$ represents the factor loading linked to the dichotomous variable $i$ within factor $j$. $D_j$ denotes the total number of dichotomous variables in factor $j$. Finally, $SFL_j$ is the sum of factor loadings for factor $j$. This index enables the comparison of municipal portals based on factors identified in the factor analysis. A MIFL value of 1 signifies that the municipal portal analyzed contains all the required information, whereas a value of 0 indicates the absence of information related to that factor.

Subsequently, utilizing MIFL along with the explained variance of each factor ($EV_j$), we formulate MDOI as per Eq. (2), facilitating a comprehensive comparison among the various municipal portals nationwide. This method ensures that each factor's importance is proportional to its contribution to the overall explained variance, resulting in a fair and comprehensive measurement. The index formula considers both the factor loadings ($FL_{ij}$) and the statistical relevance of each factor in explaining the presence of municipal digital services. An MDOI value of 1 indicates that the municipal portal provides information for all necessary aspects, whereas an MDOI value of 0 signifies that the municipal portal does not offer any required information.

$$MDOI_k = \sum_{j=1}^{15} MIFLjk * EV_j \ \forall \ k \in \{1, \ 2, \ \ldots, \ 344\} \tag{2}$$

**Table 2 Factors, eigenvalues and explained variance.**

| Factor | Eigenvalues | Explained variance | Cumulative explained variance |
|---|---|---|---|
| 1. Municipal Unit and Official Communication | 9.40 | 11.0% | 11.0% |
| 2. Integration Efforts | 7.71 | 9.0% | 19.0% |
| 3. Transparency Measures | 7.16 | 8.0% | 28.0% |
| 4. Educational and Health Resources | 6.95 | 8.0% | 35.0% |
| 5. Lobbying Regulations | 5.84 | 7.0% | 42.0% |
| 6. Communal Council for Civil Society Organizations (COSOC) | 4.96 | 6.0% | 48.0% |
| 7. Benefits, Responsibilities, and Fees | 4.42 | 5.0% | 53.0% |
| 8. Informative Content Distribution | 4.23 | 5.0% | 58.0% |
| 9. Community Project Initiatives | 3.47 | 4.0% | 62.0% |
| 10. Online Transaction Processes | 3.37 | 4.0% | 65.0% |
| 11. Ethical standards | 2.71 | 3.0% | 68.0% |
| 12. Citizen Engagement and Consultation | 2.71 | 3.0% | 72.0% |
| 13. Orientation Services | 2.47 | 3.0% | 74.0% |
| 14. Environmental Initiatives | 2.47 | 3.0% | 77.0% |
| 15. Security and Reliability Standards | 2.08 | 2.0% | 79.0% |

Finally, a multiple linear regression (MLR) model is implemented with the purpose of predicting the value of the MDOI through various independent variables of interest.

# RESULTS AND DISCUSSION

## Factor analysis

The reliability of the instrument was assessed with Cronbach's alpha coefficient, which exceeded the standard of excellence with a value above 0.90, reflecting a high internal consistency of the items and providing confidence in the consistency of the measurements. The suitability of the data for factorization was confirmed through the KMO measure, which was found to exceed the recommended threshold of 0.80, in line with guidelines established by *Lloret-Segura et al. (2014)*, thereby ensuring the relevance of factor analysis for the dataset in question.

The least squares residuals methodology, used for factor extraction, contributed to an optimized data structure by minimizing discrepancies between observed and modeled correlations. This led to the elimination of 55 variables, thus focusing on those variables that present a stronger and more representative relationship with the identified factors. EFA effectively summarizes the original 88 variables into 15 factors that explain 79% of the data variability, a percentage that far exceeds the commonly recommended threshold of 60% for cumulative explained variances. This demonstrates an effective data reduction while maintaining considerable informational richness. Table 2 summarizes the selected factors, eigenvalues, and the percentage of variation explained.

Factor Load for each variable can be seen in Table A2. The factors were further clarified based on the elements they consist of. *Municipal Unit and Official Communication* (Factor 1) gathers variables that offer insights into councilors, as well as details from the

Community Development Department (DIDECO) in charge of fostering growth and improvement within the community, such as addresses for municipal works, cabinet, and community development offices, among others. *Integration Efforts* (Factor 2) is about details regarding integration programs for women and people with disabilities on the municipal website. *Transparency Measures* (Factor 3) pertains to municipal transparency concerning the accessibility of information on the web portal regarding internal matters. Variables are linked to details found on the active transparency portal, as well as other variables indirectly linked to transparency, like information on the involvement of other entities in local projects.

*Educational and Health Resources* (Factor 4) relates to information about educational and health facilities located within the municipality. *Lobbying Regulations* (Factor 5) check if the local platform meets all the requirements stated by the law, regarding paid activities or efforts conducted by Chilean or foreign individuals or entities to support, defend, or represent specific interests or influence decisions made by authorities. At the local level, regional councilors, mayors, councilors, executive secretaries of regional councils, construction directors, and municipal secretaries are open to lobbying activities. *Communal Council for Civil Society Organizations (COSOC)* (Factor 6) pertains to details about the COSOC, which serves as a platform to participate in governmental management, policies, programs, and plans, acknowledging their right to be involved.

*Benefits, Responsibilities, and Fees* (Factor 7) concerns details about the benefits available to residents of the municipality, like the neighbor card, vaccination schedules, and social housing registration. It also involves responsibilities such as waste collection fees, and fines payments. *Informative Content Distribution* (Factor 8) consists of details about emergency activities and involvement in the community. *Community Project Initiatives* (Factor 9) offers details and descriptions of projects initiated by the municipality for the welfare of the community. *Online Transaction Processes* (Factor 10) relate to the municipality's readiness to conduct digital procedures, like applying for certificates, patents, circulation permits, and making inquiries.

*Ethical standards* (Factor 11) consist of three variables indicating whether the website includes the commune's mission, vision, and code of integrity. *Citizen Engagement and Consultation* (Factor 12) consists of indicators that assess whether the municipality provides tools to address and communicate information to its users. *Orientation Services* (Factor 13) includes variables related to the tools available on the website, aiming to assist users with accessing the *Chile Atiende* portal and obtaining essential circumstantial information. *Environmental Initiatives* (Factor 14) comprises variables that assess if the website contains information regarding environmental care within the community. *Security and Reliability Standards* (Factor 15) measure whether the municipality provides information about various mechanisms employed to enhance security in different areas, including privacy, citizen safety, emergencies, and accidents.

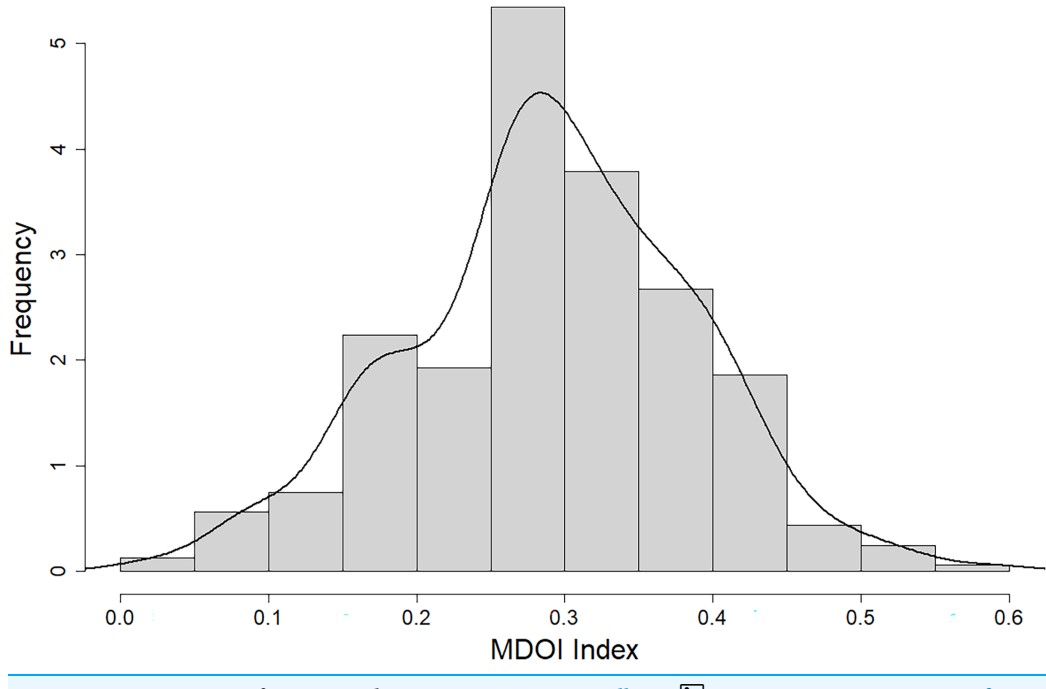

**Figure 2  Histogram of MDOI Index.**               

## Municipal Digital Offering Index

Aiming to answer the research question on the development level of the municipal digital offering in Chile (RQ1), we created the MDOI, enabling the creation of a systematic ranking that classifies Chilean municipalities according to their performance in digital offerings through their website. This classification allows for the distinction between municipalities with high levels of digital development, making information and interaction fully available for citizens through their website, and those that still have room for improvement.

Index scores range from 0–1, where 0 indicates the absence of a certain factor, and one indicates the complete presence of the elements that compose that factor.

Figure 2 shows that the MDOI index in Chilean municipalities does not surpass 0.6 and it mainly falls within the range of 0.25 to 0.4, suggesting a general low level of digital development. Three municipalities in the Metropolitan region have the highest index: Lo Barnechea (0.59), Las Condes (0.55), and Santiago (0.53), revealing an important gap and improvement opportunities for local e-government in Chile. The first two municipalities mentioned are categorized as high-income, while the third one is the central area of the city, which encompasses the historic center, the original neighborhoods of the city, and the primary government institutions.

Figure 3 illustrates the regional disparities in digital development across different areas. La Araucanía and Valparaíso regions stand out for their superior performance. The Metropolitan Region shows extreme cases, with three municipalities having the highest scores, while Calera de Tango (0.01), a rural municipality, is at the lowest end of the digital development spectrum. The figure also highlights the uneven variability of the index

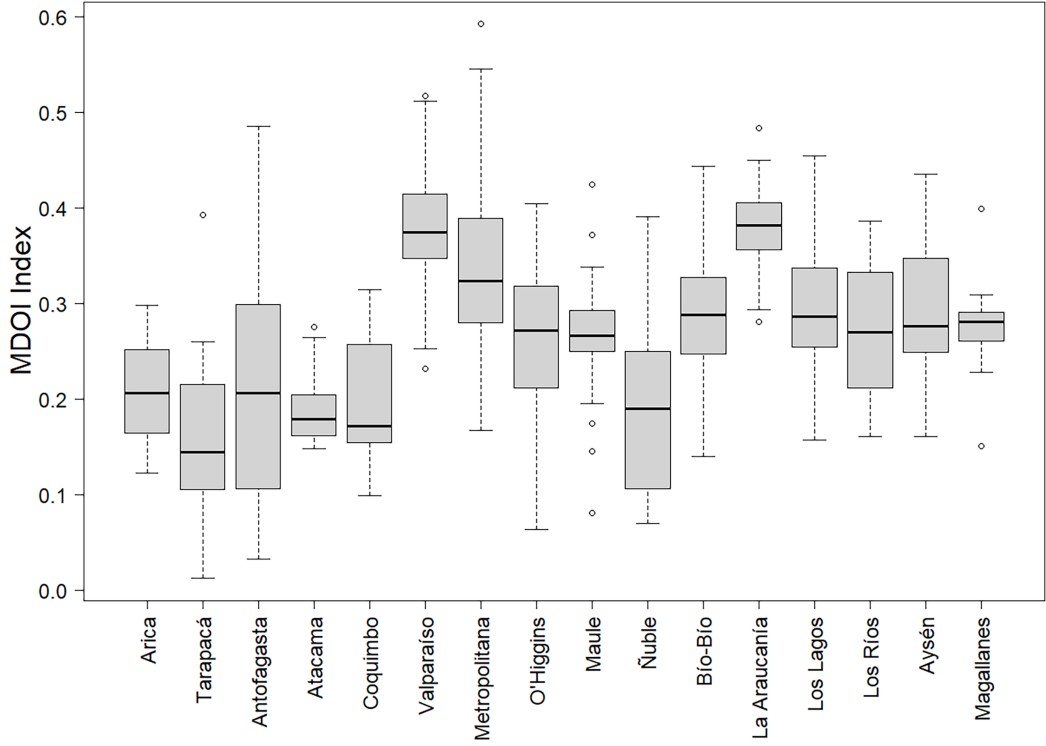

**Figure 3** **Regional variation in MDOI: boxplot by regions.**

**Table 3** **MIFL statistical summary.**

| Factor | Mean | Desv.Est. | Max | Min | % <0.3 |
|---|---|---|---|---|---|
| Municipal Unit and Official Communication | 0.72 | 0.36 | 1.00 | 0 | 20% |
| Integration Efforts | 0.15 | 0.24 | 0.79 | 0 | 75% |
| Transparency Measures | 0.74 | 0.11 | 0.77 | 0 | 2% |
| Educational and Health Resources | 0.20 | 0.24 | 0.85 | 0 | 64% |
| Lobbying Regulations | 0.47 | 0.19 | 0.58 | 0 | 14% |
| Communal Council for Civil Society Organizations (COSOC) | 0.14 | 0.18 | 0.51 | 0 | 78% |
| Benefits, Responsibilities, and Fees | 0.08 | 0.09 | 0.50 | 0 | 96% |
| Informative Content Distribution | 0.07 | 0.07 | 0.27 | 0 | 99% |
| Community Project Initiatives | 0.01 | 0.03 | 0.21 | 0 | 100% |
| Online Transaction Processes | 0.04 | 0.07 | 0.34 | 0 | 98% |
| Ethical standards | 0.00 | 0.01 | 0.06 | 0 | 100% |
| Citizen Engagement and Consultation | 0.06 | 0.09 | 0.25 | 0 | 100% |
| Orientation Services | 0.11 | 0.11 | 0.30 | 0 | 85% |
| Environmental Initiatives | 0.02 | 0.04 | 0.16 | 0 | 100% |
| Security and Reliability Standards | 0.01 | 0.03 | 0.18 | 0 | 100% |

within regions. For example, the Antofagasta and Metropolitan regions exhibit greater dispersion in digital development compared to the more uniform Magallanes and Atacama regions.

A thorough examination of factors uncovers both strengths and weaknesses in the execution of digital services. Table 3 provides a statistical overview of the MIFL. The final column indicates the percentage of municipal portals with an MIFL < 0.3, indicating subpar performance regarding the respective factor.

Areas such as *Lobby Regulations*, *Transparency Measures*, and *Municipal Unit and Official Communication* present better performance with MIFL media over 0.47, while other aspects like *Citizen Engagement and Consultation*, *Security and Reliability standards*, *Community Project Initiatives*, *Orientation services*, *Online Transaction Processes*, *Benefits, Responsibilities, and Fees* (MIFL < 0.11) need significant improvements. These poorly developed aspects are also highly frequent, given that over 85% of municipalities have an index below 0.3.

Emphasis should be placed on *Citizen Engagement and Consultation*, as none of the municipalities offer avenues for participation. Furthermore, 98% of municipalities exhibit inadequate *Online Transaction Processes*, raising concerns about digital citizenship progress (*Ribble, Bailey & Ross, 2004*; *Pade-Khene, 2018*). Despite increased online activity during the pandemic, it is plausible to speculate that most interactions with municipalities still occur in person.

## Multivariate linear regression (MLR)

To answer research question 2 on the demographic, social, and economic factors related to the municipal digital offering level (RQ2), we developed a multiple linear regression analysis incorporating the following explanatory variables:

- Log (CDI): The natural logarithm of the CDI for the year 2020. The CDI aggregates various community-level variables into three dimensions: Health and Social well-being, Economy and Resources, and Education. More details on this index can be found in *Hernández Bonivento et al. (2020)*.
- Log (Size): The natural logarithm of the area of the commune measured in square kilometers ($km^2$).
- Log (No schooling): The natural logarithm of the percentage of residents in the commune who do not have formal schooling.
- Log (Income): The natural logarithm of the total income of the commune.
- Region: Categorical variables indicating the region where the municipality is located.

The variable 'Size' was selected for its strong correlation with population size, density, and urban population percentage. A logarithmic transformation was employed for both, the dependent variable (MDOI index) and the explanatory continuous variables, to improve the model fit. This transformation served to mitigate extreme values and diminish the magnitude of fluctuations. The categorical variable 'Region' is also included to provide a comparison of the MDOI index between regions of Chile, considering the Metropolitan

**Table 4  Results of the multiple linear regression.**

| Predictor | Coefficient | White robust standar error | Sum of square from ANOVA as % to total sum of squares |
|---|---|---|---|
| Constant | **−2.54** | 0.40 | – |
| log (CDI) | **0.19** | 0.09 | 9.18 |
| log (Size) | 0.04 | 0.03 | 1.71 |
| log (No_Schooling) | **−0.31** | 0.13 | 1.27 |
| log (Income) | **0.05** | 0.02 | 1.15 |
| Arica y Parinacota | **−0.38** | 0.16 | 34.01 |
| Tarapacá | −0.93 | 0.53 | |
| Antofagasta | **−0.86** | 0.36 | |
| Atacama | **−0.53** | 0.16 | |
| Coquimbo | **−0.48** | 0.13 | |
| Valparaíso | **0.17** | 0.07 | |
| Metropolitana de Santiago | (reference category) | | |
| del Liberatdor B. O'Higgins | −0.15 | 0.10 | |
| Maule | −0.15 | 0.10 | |
| Ñuble | **−0.49** | 0.15 | |
| Bio-bio | −0.06 | 0.08 | |
| La Araucanía | **0.30** | 0.10 | |
| Los Ríos | −0.15 | 0.12 | |
| Los Lagos | −0.07 | 0.11 | |
| Aysén | −0.08 | 0.18 | |
| Magallanes y Antártica Chilena | −0.35 | 0.24 | |
| R2 | 0.46 | | |
| Breush Pagan Test (*p*-value) | 66.265 (3.799e−07) | | |
| F (*p*-value) | 13.69 (<2.2e−16) | | |

**Note:**
 The dependent variable is the logarithm of MDOI index. Significant coefficients at 5% level are indicated in bold. Regions are ordered from north to south.

area of Santiago as the reference region. The resulting estimated model and its outcomes are presented in Table 4.

The regression outcomes underscored that the model explains 46% of the index's variability. To consider the potential variance heterogeneity across regions (as shown in Fig. 3), the White robust standard errors (*White, 1980*) are employed. The parameters for the explanatory variables have the expected signs and show satisfactory statistical significance at the 5% level, except 'log(Size)', significant at the 10% level.

The ANOVA analysis divides the total sum of squares into percentages for each explanatory variable, showing that 'Region' and 'CDI' contribute significantly to explaining the variability of the MDOI index, accounting for 34.01% and 9.18% of the total variation. The analysis revealed that communes with a higher CDI present more advanced municipal portals, and larger ones tend to invest more in web development. The proportion of the population without education is negatively associated with the development of municipal portals. Additionally, municipal revenues correlate positively with the digitalization of web services.

Regarding regional coefficients, Arica y Parinacota, Antofagasta, Atacama, Coquimbo, and Ñuble exhibited negative coefficients, signaling lesser municipal digital advancement in comparison to the Metropolitan Region. Conversely, Valparaíso and La Araucanía show superior digital service provision. Regional differences raise questions about the political and administrative factors contributing to these disparities, which are not captured by the research instrument.

### Contributions, limitations, and future research directions

This study contributes to the understanding of digital development in Chilean local governments by introducing the Municipal Digital Offering Index (MDOI), a tool specifically designed to assess the digital service provision of municipalities across the country. In contrast to standardized international measures, the MDOI offers a context-sensitive approach aligned with national policy frameworks and institutional realities. Recent scholarship on e-government benchmarking (*Skargren, 2020*; *Przeybilovicz, Cunha & Ribeiro, 2023*) emphasizes the need for evaluation tools that reflect local conditions and are oriented toward enhancing public value. In line with this perspective, the MDOI captures multiple dimensions of digital public service delivery as implemented in Chile's diverse municipal settings. Using this index, the study presents a nationwide assessment based on 2022 data, encompassing all Chilean municipalities and identifying key factors associated with varying levels of digital development—such as geographic region, municipal revenues, and education levels.

While the dataset provides a valuable snapshot of Chile's municipal digital landscape, it reflects conditions as of 2022. Although no significant institutional reforms have occurred since then, some municipalities may have made further progress. As such, the results should be interpreted as a baseline rather than a current assessment. Additionally, the study does not include information on the specific digital needs of local communities, which limits the ability to fully align service provision with demand.

Future research could address these limitations by updating the dataset and incorporating political and administrative variables at the regional level. One promising avenue is to apply the MDOI in a longitudinal design, particularly before and after municipal elections, to explore how changes in political leadership—such as shifts in mayoral administrations—influence digital development. This approach could help assess whether political orientation or specific administrative agendas correlate with improvements or stagnation in digital service provision. Additionally, future studies could examine the actual use (or non-use) of these digital tools by citizens, thereby complementing the supply-side analysis with insights into demand-side dynamics. Such extensions would enrich our understanding of how to design more responsive, contextually grounded, and equitable digital policies for local governments.

## CONCLUSIONS

As a result, this study provides a detailed and comprehensive evaluation of Chile's municipal digital development landscape, shedding light on geographical disparities and pointing out priority areas for future interventions. The results inform the current

situation and establish a solid foundation for subsequent research and improvement in the provision of digital services at the municipal level.

This study has successfully answered the research questions, identifying the crucial dimensions that explain municipal digital development and establishing a hierarchy among the municipalities based on their online services provided by web portals. The results highlight the influence of the population's education level and the availability of municipal financial resources on the adoption of digital technologies.

However, it is alarming that most municipalities cannot offer efficient online procedures and citizen inquiries, underscoring the need for focused investments in digital transformation to improve management and citizen interaction. In this context, the improvement of web portals, possibly subsidized by the state, emerges as a promising strategy to standardize and accelerate the digitalization of municipal procedures. This centralized orientation and financial support should help minimize the digital divide that is rooted in variables such as municipal incomes, which are related to tax revenues based on sociodemographic disparities. On the other hand, the regional differences related to political and administrative management prompt questions about public administration. They also suggest the need for national strategies to share best practices.

Given that the current research focuses on the presence of digital content without assessing its quality, future studies are recommended that include user perception and experiences for a more comprehensive evaluation. These should be based on an updated and refined database, excluding redundant or ambiguous variables, to ensure clarity in information collection and accurately reflect the dynamics of change in municipal websites.

Ultimately, the study suggests that municipalities should embark on a path of continuous improvement to reach higher technological standards and foster inclusion, transparency, and efficiency in municipal management. These efforts will reflect a commitment to modernization and improving citizen services, aligning with international trends and the expectations of an increasingly digitalized society.

### Funding
This work was supported by FONDECYT (grant number 1250586) and FONDECYT (grant number 1230473). The funders had no role in study design, data collection and analysis, decision to publish, or preparation of the manuscript.

### Grant Disclosures
The following grant information was disclosed by the authors:
FONDECYT: 1250586, 1230473.

## Competing Interests

The authors declare that they have no competing interests.

## Author Contributions

- Carolina Busco conceived and designed the experiments, performed the experiments, analyzed the data, prepared figures and/or tables, authored or reviewed drafts of the article, and approved the final draft.
- Felipe González conceived and designed the experiments, analyzed the data, performed the computation work, authored or reviewed drafts of the article, and approved the final draft.
- Paula Farina analyzed the data, performed the computation work, authored or reviewed drafts of the article, and approved the final draft.
- Jonathan Vivas performed the experiments, analyzed the data, performed the computation work, prepared figures and/or tables, authored or reviewed drafts of the article, and approved the final draft.
- Fernanda Saavedra performed the experiments, analyzed the data, performed the computation work, prepared figures and/or tables, authored or reviewed drafts of the article, and approved the final draft.
- Lizbeth Avalos performed the experiments, analyzed the data, performed the computation work, prepared figures and/or tables, authored or reviewed drafts of the article, and approved the final draft.

## Data Availability

The code is available in the Supplemental File.

## Supplemental Information

Supplemental information for this article can be found online at http://dx.doi.org/10.7717/peerj-cs.3049#supplemental-information.

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
