# Peer review of "Introducing the municipal digital offering index for evaluating online services and addressing the digital divide"

_PeerJ Computer Science, doi:10.7717/peerj-cs.3049_

## Round 0.1 · original submission · Major Revisions

**Language Note:** The review process has identified that the English language must be improved. PeerJ can provide language editing services - please contact us at [email protected] for pricing (be sure to provide your manuscript number and title). Alternatively, you should make your own arrangements to improve the language quality and provide details in your response letter. – PeerJ Staff

Reviewer 2 ·

Basic reporting

The paper generally employs professional English throughout and fits within the scope of the journal.
Introduction: Clarification of "Digital Development": The introduction requires clarification regarding the central concept of "digital development." The current draft appears to conflate this term with related but distinct concepts such as e-government, digital government, and digital implementation. A precise definition of "digital development" as it is used within this paper is essential to establish a clear theoretical foundation.
Page 3, Line 87: Use of "Maturity Level": Exercise caution in using the term "maturity level." This concept carries a specific definition within the literature on maturity models. Given that the paper's focus does not appear to be on maturity levels and models, either ensure a precise and appropriate definition is provided and consistently applied, or consider alternative terminology.

The literature review seems somewhat outdated. To strengthen the paper's scholarly grounding, please incorporate more current references and data sources. For instance, regarding e-government comparative rankings (page 5, line 179 onwards), the latest iteration, the 19th Waseda University - IAC World Digital Government Ranking 2024 Survey - Report, should be considered. Furthermore, to enhance the development of Section 2.2, "E-Government Comparative Rankings," I suggest exploring relevant contemporary literature such as:
Skargren, F. (2020). What is the point of benchmarking e-government? An integrative and critical literature review on the phenomenon of benchmarking e-government. Information Polity, 25(1), 67-89.
Przeybilovicz, E., Cunha, M. A., & Ribeiro, M. M. (2023). Decolonizing e-government benchmarking. In Proceedings of the 24th Annual International Conference on Digital Government Research (pp. 570-582). These references should also be integrated into the discussion section to provide a more robust contextualization of the findings.

The paper is currently missing crucial sections that are standard for a research article, including a clear articulation of limitations and a summary of contributions and implications. These sections must be developed.

Experimental design

Research questions are well defined, including the clarification of the research gap.
Methods are well described, however, given that the research instrument was applied in 2022, and it is now 2025, the authors should address the temporal relevance of their data. If updating the data is not feasible, a thorough discussion of potential changes and their implications since 2022 should be included in a dedicated limitations section.

Validity of the findings

Findings are discussed and supported by data. The underlying data are provided. Conclusions are well stated and linked to original research questions.

Reviewer 3 ·

Basic reporting

All comments have been addressed in the final section.

Experimental design

All comments have been addressed in the final section.

Validity of the findings

All comments have been addressed in the final section.

Additional comments

Review Report for PeerJ Computer Science
(Introducing the municipal digital offering index for evaluating online services and addressing the digital divide)

1. Within the scope of this study, the digital service development of Chilean municipalities was evaluated using the Municipal Digital Offering Index, analyzing the digital divide, socioeconomic influences, and local digital transformation needs.

2. In the introduction, detailed explanations were made about what contemporary society is, the importance of the subject, and municipalities. The approach and information in this section seem to be sufficient for the study.

3. In the literature review section, studies related to the literature within the scope of the study were mentioned from E-Government Comparative Rankings, Legal and Regulatory Framework of E-Government in Chile, and Conceptual Evolution and Impact of E-Government. In this section, it is recommended to make a literature table arrangement, especially regarding the originality of the studies in the literature.

4. When the dataset used in the study is examined in detail, it is very compatible for the study in terms of both type and quantity. The collection of the dataset specific to the study increased the quality of the study.

5. The results obtained in the study, along with the types of metrics used, are found to be highly consistent with those reported in the existing literature, thereby supporting the validity and effectiveness of the proposed methodology

In conclusion, this study has the potential to make a very important contribution to the literature. However, attention should be paid to the above sections

---

## Round 0.2 · accepted · Accept

Both reviewers approve the revisions and therefore I'm happy to recommend acceptance.

Reviewer 2 ·

Basic reporting

The manuscript has been substantially enhanced regarding its literature references and overall structure.

Experimental design

The introduction now includes clearer definitions and clarifications, and the manuscript more clearly explains how it addresses the knowledge gap.

Validity of the findings

The findings are clearly stated, ensuring their significance is readily apparent.

Reviewer 3 ·

Basic reporting

All comments have been addressed in the final section.

Experimental design

All comments have been addressed in the final section.

Validity of the findings

All comments have been addressed in the final section.

Additional comments

Review Report for PeerJ Computer Science
(Introducing the municipal digital offering index for evaluating online services and addressing the digital divide)

Thank you for the revision. Both the responses to the reviewer comments and the corresponding changes made to the paper were generally appropriate.